# Survival Disparities in US Black Compared to White Women with Hormone Receptor Positive-HER2 Negative Breast Cancer

**DOI:** 10.3390/ijerph20042903

**Published:** 2023-02-07

**Authors:** Leann A. Lovejoy, Craig D. Shriver, Svasti Haricharan, Rachel E. Ellsworth

**Affiliations:** 1Chan Soon-Shiong Institute of Molecular Medicine at Windber, Windber, PA 15963, USA; 2Murtha Cancer Center/Research Program, Uniformed Services University of the Health Sciences and Walter Reed National Military Medical Center, Bethesda, MD 20889, USA; 3Department of Surgery, Uniformed Services University of the Health Sciences, Bethesda, MD 20889, USA; 4Cancer Center, Sanford Burnham Prebys Medical Discovery Institute, La Jolla, CA 92037, USA; 5Henry M. Jackson Foundation for the Advancement of Military Medicine, Bethesda, MD 20817, USA

**Keywords:** breast cancer, disparity, Black, hormone receptor positive/HER2 negative, biological, treatment

## Abstract

Black women in the US have significantly higher breast cancer mortality than White women. Within biomarker-defined tumor subtypes, disparate outcomes seem to be limited to women with hormone receptor positive and HER2 negative (HR+/HER2−) breast cancer, a subtype usually associated with favorable prognosis. In this review, we present data from an array of studies that demonstrate significantly higher mortality in Black compared to White women with HR+/HER2-breast cancer and contrast these data to studies from integrated healthcare systems that failed to find survival differences. Then, we describe factors, both biological and non-biological, that may contribute to disparate survival in Black women.

## 1. Introduction

In 2022, an estimated 42,250 women in the United States (US) are expected to die from breast cancer, making it the second leading cancer cause of death [1]. In Black women, breast cancer is the leading cause of cancer death, with an estimated 6800 Black women expected to die from breast cancer in 2022 [2]. Starting in 1990, mortality rates decreased significantly for White without a similar decrease for to Black women, resulting in a 41% higher mortality rate for Black compared to White women between 2015 and 2019 [2].

While the disparate survival noted above reflects differences across all breast tumors, breast cancer is a heterogeneous disease, including variable expression of estrogen receptor (ER), progesterone receptor (PR) and HER2 proteins. Hormone receptor positive (ER and/or PR positive) and HER2− negative (HR+/HER2−) breast cancer is both the most frequently diagnosed subtype (73%) in the US and is associated with the best prognosis, with a 5-year relative survival rate of 92% [3]. Common perception links breast cancer diagnoses in Black women in the US with triple negative (or ER−/PR−/HER2−) breast cancer (TNBC), but HR+/HER2− breast cancer remains the most commonly diagnosed subtype in this cohort (61%) [3]. Despite its generally favorable prognosis, a number of studies detected higher mortality rates for Black women with HR+/HER2− breast cancer [4,5,6].

Given that an estimated 22,000 Black women are diagnosed with HR+/HER2− breast cancer in the US each year, it is critical to understand the drivers of disparate outcomes in Black compared to White women. In this literature review, we present survival data from a range of studies, including those evaluating disparities within universal insurance or equal-access healthcare settings. In conjunction, we will present data on both biological and non-biological factors that may contribute to this disparity. The goal of this literature survey is to provide a comprehensive overview of the research that has been performed to date that supports and offers explanations for the survival disadvantage of Black compared to White women with HR+/HER2− breast cancer.

## 2. Materials and Methods

PubMed database (https://www.ncbi.nlm.nih.gov/pubmed) was searched for relevant articles (accessed on 2 June 2022) by two authors. Using the search terms BLACK/AFRICAN AMERICAN and BREAST CANCER (n = 3915), search criteria was further refined to include SURVIVAL (n = 1396 articles), SUBTYPE (n = 316 articles), ONCOTYPEDX (n = 13 articles) and ENDOCRINE THERAPY (n = 65 articles). Articles that included tumors with only hormone receptor status or with both HER2+ and HER2− tumors were excluded. Only articles written in English were included. A total of 82 articles were discussed in this review.

## 3. Results

### 3.1. Survival

Lund et al. published one of the earliest reports of higher mortality for Black compared to White women with HR+/HER2− breast cancer. In a group of 41 Black and 231 White women diagnosed with HR+/HER2− breast cancer in metropolitan Atlanta, the hazard ratio (HR) for all-cause mortality was 1.6 (95% CI 1.1–2.4) [7]. After adjustment for age, stage, grade, poverty index, treatment and treatment delay, the risk of all-cause mortality was no longer significantly different (HR 0.8, 95% CI 0.5–1.3). In contrast, a study by O’Brien et al., from the Carolina Breast Cancer Study (CBCS), evaluated breast cancer–specific survival (BCSS) in 246 Black and 379 White women with HR+/HER2-breast cancer and found that Black women were significantly more likely to die of disease than White women, even after adjusting for age, date and stage at diagnosis (HR 1.9, 95% CI 1.3–2.9) [5].

Since the publication by O’Brien et al. in 2010 [5], a number of other studies have validated their finding, identifying higher mortality in Black compared to White women with HR+/HER2-breast cancer (Table 1) [4,6,8,9,10,11,12,13,14,15,16]. For example, sub-analysis of Black and White women with HR+/HER2-breast cancer enrolled in a clinical trial comparing the efficacy of different taxane regimens revealed significantly worse disease-free (HR 1.58, 95% CI 1.19–2.10), overall (HR 1.49, 95% CI 1.05–2.02) and BCSS (HR 1.65, 95% CI 1.11–2.46) in Black women [13]. Ma et al. evaluated survival by biomarker-defined subgroups in women from Detroit and Los Angeles recruited into the Women’s CARE study [4]. Using data from 244 Black and 405 White women with HR+/HER2-breast cancer, Black women were found to have worse BCSS than White women (HR 1.52, 95% CI 1.01–2.28). Further stratification found that the difference was predominantly found in women aged 50–64 years and with p53 negative tumors. This difference was, however, no longer significant when adjusted for stage at diagnosis. Collin et al. also evaluated survival by subtype in a cohort of women from Atlanta and found that breast cancer mortality was more than two-times higher in Black compared to White women with HR+/HER2-breast cancer [9]. In a recent study of women with ER+/HER2− women with negative lymph nodes enrolled in the TAILORx trial, Albain et al. found that despite having no significant difference in use of chemotherapy or endocrine therapy and receipt of standard therapies, Black women had significantly worse invasive disease-free survival (HR 1.28, 95% CI 1.05–1.57) [8]. In one of the largest studies, using data from 18 Surveillance, Epidemiology and End Results (SEER) registries, Lorona et al. evaluated breast cancer mortality risk by age at diagnosis (<50 or ≥50 years of age) and stage [11]. Black women <50 years of age had higher risks of mortality for all stages of breast cancer. For Black women ≥50 years of age, only those with stage IV HR+/HER2− breast tumors had significantly higher risk of mortality than their White counterparts. In contrast, Zhou et al. evaluated rates of recurrence in 46,027 women ≥65 years of age with HR+/HER2−, stage I-III breast cancer and found that the five-year cumulative incidence of recurrence was higher in Black compared to White women (21.2%, 95% CI 19.4, 23.1) compared to White women (16.1%, 95% CI 15.7, 16.5) [17]. Variability in outcome differences amongst these studies has been attributed to factors, such as poverty index and access to care.

While these studies demonstrate higher risk of mortality for Black compared to White women with HR+/HER2-breast cancer, other studies have found non-disparate outcomes. In 2016, Costantino et al. evaluated whether there were significant survival differences between Black (n = 90) and White women (n = 308) diagnosed with HR+/HER2-breast cancer who were diagnosed at an equal-access military healthcare facility [18]. With an average length of follow-up of 8 years, no significant differences were detected for either progression-free or overall survival between populations. Similarly, Haque et al. evaluated survival differences between Black and White women enrolled in the Kaiser Permanente Southern California integrated health care delivery system [19]. Although crude subsequent breast cancer rates were higher in Black (48/1000 person-years) compared to White (34/1000 person-years) women with HR+/HER2− tumors, the adjusted HR was not significantly different (HR 1.10, 95% CI 0.84–1.45). Given that times to recurrence and death for HR+ tumors are 5–20 years and ≥10 years, respectively, these studies may not have had sufficient follow-up time to detect survival differences in women with HR+/HER2-breast cancer; alternatively, these studies highlight the importance of provision of integrative healthcare in ameliorating the survival disadvantage of Black women in the US with HR+/HER2-breast cancer.

### 3.2. Biological Factors

A number of biological factors may contribute to higher mortality rates in Black women. For example, while HR+/HER2− tumors are defined using IHC biomarkers, levels of tumor staining may differ between populations. In conjunction, the use of additional biomarkers, such as Ki67, may identify differences in cellular proliferation between populations. Evaluation of the tumor using gene expression analysis may identify molecular differences within the tumor associated with less favorable outcomes.

#### 3.2.1. ER and PR Positivity

The extent of ER and PR staining within a tumor may influence patient outcome. In 2010, the American Society of Clinical Oncology and College of American Pathologists published guidelines for determining hormone receptor status in breast tumors, with a cutoff of 1% positive cells defining ER positive status [20] with updated guidelines published in 2020 recommending that tumors with 1–10% ER positive cells should be classified as ER low positive [21]. Multiple gene expression-based studies have shown that the majority of ER low positive tumors have ER negative intrinsic subtypes (basal-like or HER2-enriched) [22,23]. Moreover, patients with ER low positive tumors had survival rates similar to those with ER negative tumors [24]. In a study of 1238 women with invasive breast cancer, Black women (8.1%) were found to be significantly (<0.001) more likely to have ER low positive tumors than White women (3.4%) [22]. In 2020, two publications evaluated ER levels in Black and White women with HR+/HER2-breast cancer. Purrington et al. evaluated ER staining levels in a cohort of 1573 women from Detroit, MI, classifying ER staining levels as weak (1–10% staining), moderate (11–50% staining) or strong (>50%) and found that Black women were significantly more likely to have both weakly staining (odds ratio (OR) 2.19, 95% CI 1.14–4.23) and moderately staining (OR 2.80, 95% CI 1.37–5.71) breast tumors comped to White women [25]. Within the CBCS, Black women were not significantly more likely to have ER low positive tumors than White women; however, in those with ER low positive tumors, Black women (38.8%) were significantly more likely to have higher risk of recurrence than White (12.5%) women [26]. White women with ER low positive tumors who underwent endocrine therapy had recurrence risks similar to those with ER positive tumors while those who did not undergo endocrine therapy had significantly worse disease-free interval (HR 4.22, 95% CI 1.75–10.23). In contrast, Black women with ER low positive tumors had worse disease-free intervals whether (HR 2.77, 95% CI 1.09–7.04) or not (HR 2.53, 95% CI 1.13–5.70) they utilized endocrine therapy. In conjunction with these studies evaluating protein levels of ER, Wright et al. found that Black women with ER+/PR+/HER2− tumors had significantly worse progression-free and overall survival than White women [27]. No significant difference was detected for women with ER+/PR−/HER2− tumors. Together, these data suggest that differences in protein expression of hormone receptors may contribute to survival disparities in Black women with HR+/HER2− breast tumors; however, it is likely that other factors are also contributory.

#### 3.2.2. KI67

Ki-67, expression levels, which correlate with cellular proliferation, have been used to further divide HR+/HER2− tumors into luminal A (HR+/HER2−/low Ki-67) and luminal B (HR+/HER2−/high Ki-67) subtypes [28]. Outcomes are significantly different between these two groups with patients with luminal A tumors having 10-year survival estimates of 70% and 15-year distant relapse rates of 27.8% compared to 54.5% and 42.9% in patients with luminal B tumors [29]. In the study by Costantino et al. [18], Black women with HR+/HER2− tumors (26%) were significantly more likely (*p* < 0.001) to have high Ki-67 staining tumors than White women (9%). Using the four biomarker classification scheme, 19% of tumors from Black women would be classified as luminal B compared to 6% from White women. Similarly, other studies suggest that HR+/HER2− breast tumors from Black women have increased cell cycle progression as reflected by cell cycle gene expression scores [30].

#### 3.2.3. Discordance between IHC Biomarkers and Intrinsic Subtype

Differences between clinical biomarkers and underlying gene expression-based subtypes may also contribute to survival differences between Black and White women with HR+/HER2-breast cancer. Patterns of tumor-based gene expression can classify breast tumors into different subtypes [31,32] with different treatment strategies [33] and sites of metastasis and prognosis associated with each [29]. In the study by Costantino et al., when gene expression-based intrinsic subtypes were determined, Black women with HR+/HER2− tumors were more likely to have tumors with the luminal B (17%) and basal-like (10%) subtypes than those from White women (8% luminal B and 5% basal-like) [18]. Similarly, in women from the CBCS with HR+/HER2-breast cancer, Black women (51.0%) were less likely to have tumors of the luminal A subtype than White women (59.9%) [34]. In addition, Black women had higher rates of luminal B (30.3%) and basal-like (6.7%) tumors compared to White women (25.5% and 4.2% luminal B and basal-like tumors, respectively). The enrichment of more aggressive subtypes in Black women with HR+/HER2-breast cancer may contribute to the survival disadvantage. Within a group of 318 women with localized HR+/HER2-breast cancer, young Black women (47%) were more likely to have non-luminal A tumor subtypes than older Black (31%), young White (10%) and older White (30%) women [35]. While HR+/HER2− non-luminal A tumors were associated with higher 10-year mortality compared to HR+/HER2− luminal A tumors, 10-year survival did not differ significantly between Black and White women with the non-luminal A tumors (HR 1.23, 95% CI 0.58–2.58). The increased frequency of non-luminal A tumors within Black women with HR+/HER2-breast cancer may contribute to disparate survival.

#### 3.2.4. Molecular Differences within Breast Tumors

In 2015, investigators from the CBCS published one of the earliest studies evaluating molecular differences by subtype in tumors from Black compared to White women [36]. The authors reported that 23 genes were differentially expressed in luminal A tumors of which expression level differences for six genes were associated with worse survival. For two of these genes, *CRYBB2* and *PSPH*, expression levels were not only different in tumor epithelium but benign stroma as well. The authors suggested that molecular differences associated with poor survival exist from the earliest stages of tumor development. In a study from the Clinical Breast Care Project (CBCP), in which gene expression profiling was performed from tumor epithelial cells of HR+/HER2− breast tumors from 57 Black and 181 White women, 10 genes were differentially expressed between populations, with the highest-fold changes detected not for *CRYBB2* and *PSPH* but for the pseudogenes *CRYBB2P1* and *PSPHP1* [18]. Follow-up studies from CBCS found associations between higher expression of *PSPH* and risk of recurrence (HR 1.76, 95% CI 1.15–2.68) [37] and that both *CRYBB2* and *CRYBB2P1* promote tumor progression in vivo [38]. In contrast, two studies from the CBCP determined that only the pseudogenes *PSPHP1* and *CRYPBB2P1* were expressed at significantly higher levels in tumors and blood from Black compared to White women [18,39]. Moreover, expression level differences were attributable to insertion/deletion polymorphisms in both genes, with no significant differences in variant frequency between cases and controls within each population and no association with survival.

Although questions remain as to whether differentially expressed genes are associated with survival differences or represent population stratification, gene expression-based prognostic signatures support less favorable outcomes for Black compared to White women with HR+/HER2-breast cancer. For example, in a cohort of 1009 Black women and 766 White women with HR+/HER2− tumors enrolled in the CBCS, the 50 gene PAM50 assay was used to predict risk of recurrence (ROR) [40]. Black women had nearly two-fold higher ROR (crude HR 1.81, 95% CI 1.34 to 2.46) compared to White women. While the five-year recurrence risk in women with low or medium ROR was not significantly different between Black and White women, in those with high ROR, 5-year standardized recurrence risk was higher in Black women (18.9%, 95% CI 8.6–29.1%) compared to White women (12.5%, 95% CI 2.0–23.0%). In conjunction, evaluation of 21 genes in the Oncotype DX (ODX) test, Black women with HR+/HER2− were significantly more likely to have high recurrence scores (RS) compared to White women (adjusted OR 1.29 CI 95% 1.16–1.42) [41]. In a study of over 227,000 women, Moore et al. found that Black women (19.1%) were significantly more likely (*p* < 0.0001) to have high RS (≥26) compared to White (14.0%), Hispanic (14.2%) and Asian American (15.6%) women [42]. Similarly, in 86,033 patients with ODX RS, Hoskins et al. found that Black women (17.7%) were significantly more likely (*p* < 0.001) to have high RS than White women (13.7%) [43]. In addition, within women with negative lymph node status, Black women had higher mortality rates within each of the RS groups: RS 0-10 HR 2.54 (95% CI 1.44–4.50), RS 11-25 HR 1.64 (95% CI 1.23 to 2.18), RS ≥ 26 HR 1.48 (95% CI 1.10–1.98). In contrast, Albain et al. found no significant differences (*p* = 0.22) in RS between Black and White women enrolled in the TAILORx Trial which included a cohort of women who were uniformly selected, staged and treated [8].

### 3.3. Non-Biological Factors

A number of non-biological factors may also contribute to survival disparities in women with HR+/HER2-breast cancer [44]. For example, the influence of socioeconomic status (SES) on survival disparities has been known for over 40 years, with a study of data from 515 White and 388 Black women diagnosed with breast cancer between 1968 and 1977 demonstrating that survival differences were not accounted for by age or stage at diagnosis; rather, survival was attributable to differences in the distribution of SES between populations [45]. One critical difference between the early studies by Lund et al. and O’Brien et al. is that Lund et al. included poverty index and treatment in their adjusted analyses, whereas O’Brien et al. did not. To this end, O’Brien et al. suggested that the less favorable prognosis in Black women may be attributed to differences in access to care and treatment. In conjunction, both differences in health insurance status and type of health care delivery may impact patient outcomes as demonstrated by studies from the health maintenance organization Kaiser Permanente Southern California (KPSC), which found that within their system, which provides integrative healthcare to their members, breast cancer outcomes were not associated with race/ethnicity [19].

#### 3.3.1. Social Determinants of Health

Social determinants of health conditions include factors, such as economic stability, education and healthcare access and quality, neighborhood and built environment and social and community context (Social Determinants of Health—Healthy People 2030|https://health.gov/healthypeople/priority-areas/social-determinants-health, accessed on 29 January 2023). Although social determinants have been associated with survival disparities in Black women with breast cancer [46], few studies have evaluated the contribution of social determinants to survival disparities in Black women with HR+/HER2-breast cancer. Parise and Caggiano evaluated the effect of SES as measured by factors including education, employment, median household income and property values, on survival in breast cancer subtypes in a cohort of 143,184 patients from the California Cancer Registry [47]. Black women with ER+/PR+/HER2− stage 2 breast cancer had increased risk for mortality compared to White women (HR 1.51, 95% CI 1.28–1.78); adjustment for socioeconomic status (SES) reduced this risk (HR 1.32, 95% CI 1.12–1.56). Similarly, the risk for Black women with stage III ER+/PR−/HER2− breast cancer (HR 1.56 95% CI 1.13–2.17) was reduced by 7% when SES was included in the models. Sadigh et al. investigated the role of insurance status and neighborhood deprivation in women with HR+/HER2-breast cancer enrolled in the TAILORx Trial [12]. While patients with Medicare or Medicaid and those living in neighborhoods with the highest neighborhood deprivation index (NDI) had shorter overall survival than those with private insurance or living in neighborhoods with the lowest NDI, Black women had significantly shorter relapse-free interval (HR 1.39, 95% CI, 1.05–1.84) and overall survival (HR, 1.49, 95% CI, 1.10–2.99) than White women even after adjusting for neighborhood deprivation index, insurance coverage, clinicopathologic characteristics and early discontinuation of endocrine therapy. Jemal et al. evaluated factors contributing to survival disparities in women aged 18–64 years with stage I-III breast cancer; although HER2 status was not reported, excess risk of death in Black compared to White women was reduced from 105.1% to 24.9% when matched for demographics, comorbidities, insurance, tumor characteristics and treatment [48]. The authors found that differences in insurance status accounted for 37% of excess mortality. These studies suggest that social determinants contribute to but do not fully explain breast cancer survival disparities between Black and White women with HR+/HER2-breast cancer. Furthermore, there remain significant gaps in knowledge regarding how these SES variables impact the somatic epigenetic and transcriptomic molecular biology of HR+/HER2-breast cancers, and thereby outcome.

#### 3.3.2. Treatment

##### Oncotype Dx Uptake and Chemotherapy

In the 2010 study from O’Brien et al., the authors suggested that disparate survival in women with HR+/HER2-breast cancer may be attributable to differences in access to care and treatment [5]. ODX is a gene expression-based assay that measures risk of recurrence in women with HR+/HER2-breast cancer and allows women with low-risk scores to avoid chemotherapy. A number of studies have investigated the use of the ODX test in Black compared to White women (Table 2), many of which found lower test uptake in Black women [42,49,50,51,52,53,54,55]. For example, in a study of women eligible for testing in 2010–2012, Black women were significantly less likely to have received ODX results compared to White women (OR 0.732 95% CI 0.702 to 0.763) [51]. Similarly, in a cohort of 227,259 women diagnosed with ER+, early stage, node negative breast cancer, 32.8% of Black women and 36.7% White women received ODX test results; after adjusting for demographics, clinical characteristics and access-to-care, Black women were less likely to receive test results (rate ratio 0.87, 95% CI 0.85–0.88) [42]. In 125,288 women diagnosed with HR+/HER2− node negative disease, Black women were significantly more likely to have ODX omitted from their clinical care (OR 1.25, 95% CI 1.19–1.31) [52].

Compliance with national guidelines to omit chemotherapy in low-risk women and use chemotherapy in high-risk women varied within different studies [41,51,52,56,57,58]. For example, a study by Kozick et al. found no significant differences in compliance with treatment guidelines for Black compared to White women (omission of chemotherapy in low risk women: OR 0.908, 95% CI 0.746 to 1.106; use of chemotherapy in high-risk women: OR 1.305, 95% CI 0.887 to 1.920) [51]. A study of test-eligible women from the state of Georgia 2010–2014 also found that compliance with chemotherapy guidelines was not significantly different [57]. In contrast, Han et al. found that in women from the SEER database with high RS, for whom chemotherapy is recommended, Black women were significantly less likely to use chemotherapy (aOR 0.76 95% CI 0.62–0.94) [41]. Similarly, in women from the National Cancer Database with high risk of recurrence scores, Bilani et al. found that Black women were significantly more likely to refuse chemotherapy than their White counterparts (OR: 1.20, 95% CI 1.07–1.36) [56]. Press et al. also investigated chemotherapy use using data from the National Cancer database and found that in women who underwent ODX testing, chemotherapy use did not differ significantly between Black and White women, with most women being guideline compliant [52]. In women who did not undergo testing, however, Black women were significantly more likely to undergo chemotherapy than White women (OR 1.23; 95% CI 1.11–1.37). The authors suggest that unequal use of ODX testing may result in disparate treatment, with Black women at risk of receiving non-beneficial chemotherapy and subsequent toxicities. Recently, Jung et al. investigated outcomes within patients who did and did not undergo chemotherapy [58]. Breast cancer- specific mortality (BCSM) was significantly lower for White women who underwent chemotherapy compared to those who did not (aHR 0.734, 95% CI 0.588–0.917). In contrast, BCSM did not differ significantly in Black women who did compared to those who did not undergo chemotherapy (aHR 0.748, 95% CI 0.428–1.307). The authors suggested that ODX testing may be of limited value in Black women. Since ODX testing (like most other breast cancer-ER−related prognostic/predictive tests) was developed based on data predominantly from breast cancer in White women, a revisiting of the accuracy of these tests in non-White breast cancer patients is warranted.

##### Endocrine Therapy

In addition to disparities in use of and response to ODX results, women with HR+/HER2-breast cancer are eligible for endocrine therapy as part of their treatment regimens. Delays in the initiation of endocrine therapy have been associated with decreased survival: in a study of 144,103 women, Fu et al. found that time to adjuvant hormone therapy of >150 days was associated with decreased survival (HR 1.31, 95% CI 1.26–1.35) [59]. Within this study, Black women were at increased risk for initiating endocrine therapy >150 days (OR 1.66, 95% CI 1.55–1.77). In a study by Lee et al., delayed initiation was defined as starting adjuvant endocrine therapy >12 months after diagnosis [60]. Black women were significantly more likely to have delayed initiation (aOR 1.61, 95% CI 1.52–1.70). Similarly, Reeder-Hayes et al. found that Black women in North Carolina were 17% less likely to initiate endocrine therapy within 12 months of diagnosis [61]. In women who underwent chemotherapy, Black women were less likely to initiate endocrine therapy within 12 months (aHR 0.67, 95% CI 0.56–0.80); in contrast, no difference was detected in women who did not undergo chemotherapy (aHR 0.96, 95% CI 0.76–1.21). Of note, studies that included women with insurance either through the Kaiser Permanente integrated healthcare system or Medicare found that Black women were still less likely to initiate endocrine therapy within 12 months of diagnosis [62,63,64] (Table 3). Thus, having insurance may not be the only barrier to endocrine therapy.

Suboptimal use of endocrine therapy, including premature discontinuation and incomplete adherence, may also impact patient response. For example, Hershman et al. found hazard ratios of 1.26 and 1.49 in patients who discontinued or were not adherent with adjuvant hormonal therapy [65]. Evaluation of endocrine therapy use within the Women’s Hormonal Initiation and Persistence study found that Black women were significantly less likely to be adherent than White women (OR, 0.43, 95% CI 0.27–0.67) [66]. Heiney et al. measured adherence using the medical possession ration (MPR) or the interval between refills [67]. The MRP for White women (95.7%) was significantly higher (*p* = 0.02) than that for Black women (93.4%). Farias et al. evaluated endocrine therapy use in women from Texas with Medicaid in two studies. In the first, which included 1240 women diagnosed with breast cancer 2000–2007, Black women were less likely to be adherent (OR 0.62, 95% CI 0.44–0.87) than White women, but discontinuation was not significantly different [68]. Similarly, in the second study, which included 1497 women with Medicaid who were diagnosed 2000–2008, Black women were significantly less likely to be adherent for three years (OR: 0.45, 95% CI 0.28–0.73) [69]. Similarly, a study of 1925 women, all who had insurance through health maintenance organizations, found that Black women were less likely to be 80% adherent (OR 0.72, 95% CI 0.57–0.90) and to have a medication gap of <10 days (OR 0.65, 95%CI 0.54–0.79) than White women [70]. A study from the CBCS evaluated three challenges associated with adherence with endocrine therapy: difficulty in developing medication-taking behavior (habit), high perceived side effects/medication safety (tradeoffs) and cost/accessibility (resource barrier) [71]. Black women were more likely to report barriers for each of these factors- habit: aRR 1.29, 95% CI 1.09–1.53, tradeoffs: aRR 1.32 95% CI 1.09–1.60 and resources: aRR 1.65, 95% CI 1.18–2.30). Using insurance claims data for women diagnosed with breast cancer 2007–2011, Hershman et al. found that both adherence (OR 0.76; 95% CI, 0.55 to 0.88) and discontinuation (OR 1.16, 95% CI 1.02–1.32) of endocrine therapy were higher for Black women [72]. Adjustment for net worth reduced the odds of non-adherence by 19% and, when evaluated by net worth, adherence was significantly lower in Black women only in the low net worth group. Studies by Farias et al. and Biggers et al. also found that differences in adherence and discontinuation were reduced when adjusted for financial factors such as subsidies and out-of-pocket costs [73,74].

## 4. Discussion

Multiple publications have reported less favorable survival in Black compared to White women with HR+/HER2-breast cancer in the US. While some studies have suggested that these difference may be reflective of differences in care, especially adherence to endocrine therapy [5,9], other studies found disparate outcomes even when standardized treatment was provided and/or compliance with 5-year of endocrine therapy was achieved [8,13]. In contrast, the two studies that included women treated within Kaiser Permanente’s integrated healthcare system or within an equal-access military treatment facility of the Department of Defense found no significant difference in survival between Black and White women with HR+/HER2-breast cancer [18,19].

Studies that included molecular characterization of the tumor component beyond ER, PR and HER2 status suggest that survival disparities are restricted to specific subsets of HR+/HER2− tumors. Three studies that evaluated intrinsic subtypes in HR+/HER2− tumors found that Black women were significantly more likely to have tumors that were non-luminal A subtypes [18,34,35]. Importantly, while non-luminal A tumors were not associated with increased mortality in Black compared to White women, these tumors did have less favorable outcomes than the luminal A subtype [35]. The higher frequency of these tumors within Black women may be contributing to higher mortality. Moreover, several studies have found higher gene expression-based risk scores in HR+/HER2− tumors from Black women [40,41,42,43] and, within the high-risk group, Black women had higher 5-year recurrence risk than White women [40]. These data highlight the importance of providing appropriate adjuvant treatment in the highest risk patients for reducing disparities within women with HR+/HER2-breast cancer.

The importance of providing chemotherapy and endocrine therapy to the highest-risk women highlights the interplay between biologic and non-biologic factors in driving disparate outcomes in Black compared to White women. Black women in the US are less likely to undergo molecular risk assessment [42,49,50,51,52,53,54,55]. This represents a lost opportunity not only to provide adjuvant treatment to those who would most benefit but to prevent significant treatment side effects in those who will not derive benefit from chemotherapy. Efforts to increase testing and use of chemotherapy and endocrine therapy are essential to improve survival in Black women.

A number of studies have been described in this review. However, additional studies are needed to fully understand why Black women with HR+/HER2-breast cancer have higher mortality rates than White women. To our knowledge, no systematic reviews or meta-analyses have been published that focus on HR+/HER2-breast cancer disparities. These types of studies will be invaluable not only in determining the extent of mortality differences across a range of US populations with varying types of health insurance and healthcare systems, but may identify the factors, including biological differences and provision of healthcare, that contribute to outcome differences. Future studies should also include additional evaluation of somatic gene expression or mutational frequencies in common oncogenes and tumors suppressors focusing on Black HR+/HER2-breast cancer. Existing studies and available public datasets either consider all breast cancer subtypes together preventing nuanced understanding of underlying molecular biology or have small sample sizes (<50 patients) which do not provide sufficient power for agnostic analyses. For example, the tumor microenvironment performs a critical role in tumorigenesis and differences in the microenvironment from Black compared to White women, such as higher vessel density, increased macrophage recruitment and cytokine levels, may lead to increased growth, angiogenesis, metastasis and therapy resistance [75]. While a number of studies have evaluated molecular differences in the tumor microenvironment of Black compared to White women with all subtypes of breast cancer or triple negative breast cancer [76,77,78], to our knowledge, no studies have investigated the role of the tumor microenvironment in disparate survival of women with HR+/HER2-breast cancer. In conjunction, a number of lifestyle behaviors and environmental exposures, such as higher BMI and exposure to endocrine-disrupting chemicals, have been detected in Black women [79]. While these factors may have been linked to increased risk of breast cancer, future studies are needed to determine whether these modifiable and non-modifiable factors affect somatic molecular biology of, and survival in, women with HR+/HER2-breast cancer. Emerging studies in cancer and other diseases (e.g., cardiovascular disease) suggest that socioeconomic status and systemic racism impact somatic molecular biology in ways that can be critical to tumor development and progression [80,81]. These studies suggest a new avenue of research bridging the gap between environmental and biological factors in breast cancer outcome.

It must be noted that the majority of the studies described here rely on clinician/researcher-assigned or patient-described race. As a social construct, dichotomizing women into Black or White racial groups may underestimate the impact of important social factors such as poverty and education on cancer outcomes. In addition, while the majority of people correctly self-report their race/ethnicity to major population groups, self-description cannot accurately predict the extent of admixture within an individual [82]. Future studies that stratify patients by genetic ancestry may identify additional factors, such molecular differences within the tumor or microenvironment or response to treatment, which contribute to outcome differences between Black and white women.

## 5. Conclusions

In conclusion, these data suggest that increased frequency of non-luminal A/high risk of recurrence breast tumors coupled with suboptimal provision of prognostic tests and adjuvant treatment contribute significantly to the higher mortality rates in Black compared to White women with breast cancer. However, additional studies are needed to identify other factors, both biological and non-biological, associated with disparate outcomes.

## Figures and Tables

**Table 1 ijerph-20-02903-t001:** Studies supporting survival differences in women with HR+/HER2-breast cancer.

Study	Patient Number	Population Description	Median Follow-Up	Survival Measure	Risk
O’Brien et al. [5]	246 Black, 379 White	Carolina Breast Cancer Study	9.0 years	BCSS ^a^	HR ^b^ 1.9 (95% CI 1.3–2.8)
Sparano et al. [13]	176 Black, 2803 White	Randomized phase III trial	95 months	BCSS	HR 1.65, 95% CI 1.11–2.46
Ma et al. [4]	244 Black, 405 White	Women’s CARE study	10 years	BCSS	HR 1.52, 95% CI 1.01–2.28)
Warner et al. [6]	365 Black, 6763 White	NCCN network centers	6.2 years	BCSS	HR 1.76, 95% CI 1.09–2.85
Tao et al. [14]	4813 Black, 59,341 White	California Cancer Registry	3.5 years	BCSS	HR 1.27, 95% CI 1.12–1.43
Vidal et al. [15]	521 Black, 1326 White	Memphis, TN	29.9 months	All-cause	HR 1.87, 95% CI 1.33–2.62
Collin et al. [9]	2074 Black, 3511 White	Metropolitan Atlanta	3.5 years	BCSS	HR 2.43, 95% CI 1.99–2.97
Zhao et al. [16]	613 Black, 1062 White	Chicago Multiethnic Epidemiologic Breast Cancer Cohort	6.9 years	BCSS	HR 2.37, 95% CI 1.60–3.50)
Lorona et al. [11]	20,152 Black, 148,745 White	SEER database	34 months	BCSS	HR varies by stage and age group
Zhou et al. [17]	2763 Black, 38,951 White	SEER−Medicare Linked Database	7 years	Breast cancer recurrence	Subdistribution HR 1.27, 95% CI 1.15–1.40
Albain et al. [8]	693 Black, 8189 White	TAILORx Trial	90 months	DFS ^c^	HR 1.28, 95% CI 1.05–1.57
Sadigh et al. [12]	693 Black, 8189 White	TAILORx Trial	96 months	RFS ^d^	HR 1.39, 95% CI 1.05–1.84
Du [10]	27,279 Black, 211,344 White	SEER database		BCSS	HR 1.21, 95% CI 1.06–1.37

^a^ BCSS = breast cancer-specific survival; ^b^ HR = hazard ratio; ^c^ DFS = disease-free survival; ^d^ RFS = recurrence-free survival.

**Table 2 ijerph-20-02903-t002:** Studies evaluating use of Oncotype DX testing in Black compared to White women with HR+/HER2-breast cancer.

Study	Time Period	Population Description	Risk
Cress et al. [49]	2008–2010	California Cancer Registry	OR 0.73, 95% CI 0.62–0.86
Roberts et al. [54]	2008–2014	Carolina Breast Cancer Study	aRR ^a^, 0.54, 95% CI 0.35 to 0.84
Ricks-Santi and McDonald [53]	2009–2012	Virginia Tumor Registry	Test uptake: 5.1% of Black and 11.7% of White women
Davis et al. [50]	2011–2013	Connecticut Tumor Registry	OR 0.64, 95% CI 0.47–0.88
Press et al. [52]	2010–2014	National Cancer Database	Omission of ODX: OR 1.25, 95% CI 1.19–1.31
Kozick et al. [51]	2010–2012	National Cancer Database	OR 0.732 95% CI 0.702 to 0.763
Collin et al. [56]	2010–2014	Georgia Cancer Registry	Test uptake: 47% of Black and 48% of White women
Zhang et al. [55]	2004–2015	SEER registries	aOR 0.90, 95% CI 0.82–0.99 ^b^ aOR 0.71, 95% CI 0.65–0.85 ^c^
Moore et al. [42]	2010–2014	National Cancer Database	Rate ratio 0.87, 95% CI 0.85–0.88

^a^ adjusted relative risk; ^b^ adjusted odds ratio in women with no positive lymph nodes; ^c^ adjusted odds ratio in women with 1–3 positive lymph nodes.

**Table 3 ijerph-20-02903-t003:** Studies evaluating initiation, compliance and adherence with long-term endocrine therapy use.

Study	Participants	Population Description	Parameter	Risk
Initiation				
Lee et al. [60]	391,594	National Cancer Database	>12 months	aOR 1.61, 95% CI 1.52–1.70
Fu et al. [59]	144,103	National Cancer Database	>150 days	OR 1.66, 95% CI 1.55–1.77
Reeder-Hayes et al. [61]	2640	North Carolina Central Cancer Registry	>12 months	aRR 0.83, 95% CI 0.74–0.93
Camacho et al. [63]	18,054	SEER-Medicare Database	>12 months	Black 74%, White 77%, *p* = 0.023
Bowles et al. [64]	7777	Kaiser Permanente	>12 months	RR 0.93, 95% CI 0.87–1.00
Farias et al. [62]	12,198	SEER-Medicare Database	>12 months	OR 0.25, 95% CI 0.10–0.62
Adherence and compliance				
Sheppard et al. [65]	1925	Health Maintenance Organizations		OR 0.72, 95%CI 0.57–0.90
Farias et al. [66]	1240	Texas Cancer Registry-Medicaid		OR 0.62, 95% CI 0.44–0.87
Farias et al. [67]	1497	Texas Cancer Registry-Medicaid		OR: 0.45, 95% CI 0.28–0.73
Camacho et al. [63]	18,054	SEER-Medicare Database		Black 74%, White 74%
Heiney et al. [68]	1532	A Geospatial Investigation of Breast Cancer		MDR Black 0.934 MDR White 0.957
Sheppard et al. [69]	570	Women’s Hormonal Initiation and Persistence		OR, 0.43, 95% CI 0.27–0.67

## Data Availability

Not applicable.

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
