# Peer review of "Survival Disparities in US Black Compared to White Women with Hormone Receptor Positive-HER2 Negative Breast Cancer"

_ijerph, 2023, doi:10.3390/ijerph20042903_

Round 1

Reviewer 1 Report

This is a review article detailing survival disparities in US Black versus White patients with ER pos/HER 2 neg breast cancer. Data detailing potential biological and non-biological contributions to these despairities, as well as knowledge gaps in these areas, are elucidated.

I suggest that both Black and White be capitalized, or neither capitalized. 

Some minor edits suggested: 

Abstract, proof line 19: add "breast cancer" after HR+/HER2 -

Abstract, proof line 20: a comma is missing after "non-biological"

Introduction, proof line 31: 41% higher mortality rate -- higher than what? higher than White? Or higher than for Black in previous time period? 

Results, proof lines 65-66: Articles are referred to by author name (Lund, O'Brien), which were not used earlier. Please use the author names above if you will refer to the articles by author name here. 

Results: In the section on "survival" (section 3.1), some information that would be better discussed in the sections 3.2 are discussed. For example, in proof lines 65-68 it is discussed that poverty may contribute to survival differences. 

In section 3.2.3, proof lines 186-187, the sentence starting "Similarly, in women from the CBCS ..." would be better broken into more than one sentence. The wording is confusing & in some cases incorrect. 

In Discussion, proof line 440, "use exposure" should be replaced by either "use" or "exposure"

Author Response

This is a review article detailing survival disparities in US Black versus White patients with ER pos/HER 2 neg breast cancer. Data detailing potential biological and non-biological contributions to these despairities, as well as knowledge gaps in these areas, are elucidated.

I suggest that both Black and White be capitalized, or neither capitalized. 

In response to the suggestion from reviewer 1, we have capitalized the word White throughout the manuscript. 

Some minor edits suggested: 

Abstract, proof line 19: add "breast cancer" after HR+/HER2 –

Abstract, proof line 20: a comma is missing after "non-biological"

We thank reviewer 1 for pointing out these typos and have added “breast cancer” and the missing comma  to the abstract as suggested. The revised abstract now reads:

Abstract: Black women in the US have significantly higher breast cancer mortality than White women. Within biomarker-defined tumor subtypes, disparate outcomes seem to be limited to women with hormone receptor positive and HER2 negative (HR+/HER2-) breast cancer, a subtype usually associated with favorable prognosis. In this review, we present data from an array of studies that demonstrate significantly higher mortality in Black compared to White women with HR+/HER2- breast cancer and contrast these data to studies from integrated healthcare systems that failed to find survival differences. We then describe factors, both biological and non-biological, that may contribute to disparate survival in Black women.

Introduction, proof line 31: 41% higher mortality rate -- higher than what? higher than White? Or higher than for Black in previous time period? 

We thank reviewer 1 for this suggestion and have revised the last sentence of the first paragraph of the introduction so it now reads:

Starting in 1990, mortality rates decreased significantly for White without a similar decrease for to Black women, resulting in a 41% higher mortality rate for Black compared to White women between 2015 and 2019 [2].

Results, proof lines 65-66: Articles are referred to by author name (Lund, O'Brien), which were not used earlier. Please use the author names above if you will refer to the articles by author name here. 

In response to the suggestion by reviewer 1, we have revised the first paragraph of the results section to include the first author names of the two studies described.  This section now reads:

 Lund et al. published one of the earliest reports of higher mortality for Black compared to White women with HR+/HER2- breast cancer. In a group of 41 Black and 231 White women diagnosed with HR+/HER2- breast cancer in metropolitan Atlanta, the hazard ratio (HR) for all-cause mortality was 1.6 (95% CI 1.1-2.4) [7] . After adjustment for age, stage, grade, poverty index, treatment and treatment delay, however, the risk of all-cause mortality was no longer significantly different (HR 0.8, 95% CI 0.5-1.3). In contrast, a study by O’Brien et al., from the Carolina Breast Cancer Study (CBCS), evaluated breast cancer-specific survival in 246 Black and 379 White women with HR+/HER2- breast cancer and found that Black women were significantly more likely to die of disease than White women, even after adjusting for age, date and stage at diagnosis (HR 1.9, 95% CI 1.3-2.9) [5].

Results: In the section on "survival" (section 3.1), some information that would be better discussed in the sections 3.2 are discussed. For example, in proof lines 65-68 it is discussed that poverty may contribute to survival differences. 

We thank reviewer 1 for this suggestion.  In response, we have removed the discussion of factors associated with variable survival from section 3.1, instead including a new sentence at the end of the second paragraph of 3.1 that reads:

Variability in outcome differences amongst these studies has been attributed to factors such as poverty index and access to care.

We have also moved the sentences re: differences between the Lund and O’Brien studies to section 3.1 which now reads:

A number of non-biological factors may also contribute to survival disparities in women with HR+/HER2- breast cancer [44]. For example, the influence of socioeconomic status (SES) on survival disparities has been known for over 40 years, with a study of data from 515 White and 388 Black women diagnosed with breast cancer between 1968 and 1977 demonstrating that survival differences were not accounted for by age or stage at diagnosis; but rather, were attributable to differences in the distribution of SES between populations [45]. One critical difference between the early studies by Lund et al. and O’Brien et al. is thatLund et al. included poverty index and treatment in their adjusted analyses whereas O’Brien et al. did not. To this end, O’Brien et al. suggested that the less favorable prognosis in Black women may be attributed to differences in access to care and treatment.  In conjunction, both differences in health insurance status as well as type of health care delivery may impact patient outcomes as demonstrated by studies from the health maintenance organization Kaiser Permanente Southern California (KPSC), which found that within their system, which provides integrative healthcare to their members, breast cancer outcomes were not associated with race/ethnicity [19].

In section 3.2.3, proof lines 186-187, the sentence starting "Similarly, in women from the CBCS ..." would be better broken into more than one sentence. The wording is confusing & in some cases incorrect. 

We thank reviewer 1 for this suggestion.  In response, we have separated this sentence in two.  The sentences now read:

Similarly, in women from the CBCS with HR+/HER2- breast cancer, Black women (51.0%) were less likely to have tumors of the luminal A subtype than White women (59.9%) [34]. In addition, Black women had higher rates of luminal B (30.3%) and basal-like (6.7%) tumors compared to White women (25.5% and 4.2% luminal B and basal-like tumors, respectively).

In Discussion, proof line 440, "use exposure" should be replaced by either "use" or "exposure"

We thank reviewer 1 for noting this typo and have deleted the word use so the sentence now reads:

In conjunction, a number of lifestyle behaviors and environmental exposures, such as higher BMI and exposure to endocrine-disrupting chemicals, have been detected in Black women [79].

Reviewer 2 Report

Overall conceptually it is a good manuscript and important topic. However, the methodology used is not high quality or standard for reviews. There are specific types of  reviews and should be conducted in accordance with PRISMA guidelines.

Introduction- states “in this review” but doesn’t specify what kind of review. Is this  a systematic review? A scoping review? Etc.

Materials and  methods- why  only pubmed search and not other databases? Similarly to above, if this was supposed to be any type of systematic review, the appropriate methodology for  such is not stated here. Also the search search terms were really broad, were these all the search terms used?

Results:

Majority of the results section reads like a  lengthly discussion rather than a standard concise results section

Line 165- k-67 but should be ki-67

Author Response

Overall conceptually it is a good manuscript and important topic. However, the methodology used is not high quality or standard for reviews. There are specific types of  reviews and should be conducted in accordance with PRISMA guidelines.

Introduction- states “in this review” but doesn’t specify what kind of review. Is this  a systematic review? A scoping review? Etc.

We thank reviewer 2 for this comment.  We have since specified that this was a literature review in the second sentence of the last paragraph of the introduction.  This sentence now reads:

In this literature review, we will present survival data from a range of studies, including those evaluating disparities within universal insurance or equal-access healthcare settings.

Materials and  methods- why  only pubmed search and not other databases? Similarly to above, if this was supposed to be any type of systematic review, the appropriate methodology for  such is not stated here. Also the search search terms were really broad, were these all the search terms used?

We have included more details in the methods section which now reads:

PubMed database (https://www.ncbi.nlm.nih.gov/pubmed) was searched for relevant articles (6/2022) by two authors. Using the search terms BLACK/AFRICAN AMERICAN and BREAST CANCER (n = 3915), search criteria was further refined to include SURVIVAL (n=1396 articles), SUBTYPE (n=316 articles), ONCOTYPEDX (n=13 articles) and ENDOCRINE THERAPY (n=65 articles). Articles that included tumors with only hormone receptor status or with both HER2+ and HER2- tumors were excluded. Only articles written in English were included. A total of 82 articles were included in this review.

Results:

Majority of the results section reads like a  lengthly discussion rather than a standard concise results section

We regret that we were not clear about the intent of this review; leading reviewer 2 to wonder if this was a systematic or scooping review.  We have now clarified that this was a literature review, the goal of which is to provide a comprehensive description of research into biological and non-biological factors associated with disparate survival.  We believe we have achieved this goal.  We have revised the results section in accordance with suggestions from the other reviewers. 

Line 165- k-67 but should be ki-67

We thank reviewer 2 for noting this typo and have made the correction.  The sentence now reads:

Ki-67, expression levels, which correlate with cellular proliferation, have been used to further divide HR+/HER2- tumors into luminal A (HR+/HER2-/low Ki-67) and luminal B (HR+/HER2-/high Ki-67) subtypes [28].

Reviewer 3 Report

This is an organized, comprehensive and well-written review describing the current state of knowledge regarding the disparities in breast cancer survival seen between non-Hispanic black and non-Hispanic white women after a diagnosis of HR+/HER2- breast cancer.  The authors highlight appropriate areas that may be driving some of the differences, including: molecular/biological, social determinants of health, and access to and/or adherence to testing/treatment.  Areas where data are sparse are also mentioned.

My only suggestion would be to add a short section to the Introduction, recognizing assigned (or self-reported) race is a social, not biological construct, and that the categorizations used in this manuscript reflect Census guidelines that capture this information.  Moving forward, ancestry may replace or augment work in this area and provide additional insight.

Author Response

This is an organized, comprehensive and well-written review describing the current state of knowledge regarding the disparities in breast cancer survival seen between non-Hispanic black and non-Hispanic white women after a diagnosis of HR+/HER2- breast cancer.  The authors highlight appropriate areas that may be driving some of the differences, including: molecular/biological, social determinants of health, and access to and/or adherence to testing/treatment.  Areas where data are sparse are also mentioned.

My only suggestion would be to add a short section to the Introduction, recognizing assigned (or self-reported) race is a social, not biological construct, and that the categorizations used in this manuscript reflect Census guidelines that capture this information.  Moving forward, ancestry may replace or augment work in this area and provide additional insight.

We thank reviewer 3 for the suggestion.  We have included a new paragraph in the discussion section (last paragraph) that reads:

It must be noted that the majority of the studies described here rely on clinician/researcher-assigned or patient-described race. As a social construct, dichotomizing women into Black or White racial groups may underestimate the impact of important social factors such as poverty and education on cancer outcomes. In addition, while the majority of people correctly self-report their race/ethnicity to major population groups, self-description cannot accurately predict the extent of admixture within an individual [82]. Future studies that stratify patients by genetic ancestry may identify additional factors, such molecular differences within the tumor or microenvironment or response to treatment, which contribute to outcome differences between Black and white women.   

Round 2

Reviewer 2 Report

It is unclear why a less  rigorous methodology of a plain literature review was chosen for this important topic.

Author Response

It is unclear why a less  rigorous methodology of a plain literature review was chosen for this important topic.

We appreciate that reviewer 2 is promoting the utility of a meta-analysis or systematic review re: disparities in HR+/HER2- breast cancer.  As Guest Editor of the special issue Effect of Differences in Access to Screening, Healthcare, and Treatment on Cancer Disparities, I chose to submit a literature review regarding biological and non-biological factors associated with survival disparities in Black and white women with HR+/HER2- breast cancer.  This literature review is quite fitting for this special issue topic and should serve as a preface to other submissions.  In addition, this literature provides a comprehensive resource for downstream studies, such as those that reviewer 2 is highlighting.